# Genomic and Transcriptomic Analysis for Identification of Genes and Interlinked Pathways Mediating Artemisinin Resistance in *Leishmania donovani*

**DOI:** 10.3390/genes11111362

**Published:** 2020-11-17

**Authors:** Sushmita Ghosh, Aditya Verma, Vinay Kumar, Dibyabhaba Pradhan, Angamuthu Selvapandiyan, Poonam Salotra, Ruchi Singh

**Affiliations:** 1ICMR-National Institute of Pathology, Safdarjung Hospital Campus, New Delhi 110029, India; sushmitaghosh1990@gmail.com (S.G.); adityaicmr@gmail.com (A.V.); vinaykr016@gmail.com (V.K.); poonamsalotra@hotmail.com (P.S.); 2JH-Institute of Molecular Medicine, Jamia Hamdard, New Delhi 110062, India; selvapandiyan@jamiahamdard.ac.in; 3ICMR-AIIMS Computational Genomics Centre, Indian Council of Medical Research, New Delhi 110029, India; dbpinfo@gmail.com

**Keywords:** *Leishmania donovani*, whole-genome sequencing (WGS), transcriptome, artemisinin drug resistance

## Abstract

Current therapy for visceral leishmaniasis (VL), compromised by drug resistance, toxicity, and high cost, demands for more effective, safer, and low-cost drugs. Artemisinin has been found to be an effectual drug alternative in experimental models of leishmaniasis. Comparative genome and transcriptome analysis of in vitro-adapted artesunate-resistant (K133AS-R) and -sensitive wild-type (K133WT) *Leishmania donovani* parasites was carried out using next-generation sequencing and single-color DNA microarray technology, respectively, to identify genes and interlinked pathways contributing to drug resistance. Whole-genome sequence analysis of K133WT vs. K133AS-R parasites revealed substantial variation among the two and identified 240 single nucleotide polymorphisms (SNPs), 237 insertion deletions (InDels), 616 copy number variations (CNVs) (377 deletions and 239 duplications), and trisomy of chromosome 12 in K133AS-R parasites. Transcriptome analysis revealed differential expression of 208 genes (fold change ≥ 2) in K133AS-R parasites. Functional categorization and analysis of modulated genes of interlinked pathways pointed out plausible adaptations in K133AS-R parasites, such as (i) a dependency on lipid and amino acid metabolism for generating energy, (ii) reduced DNA and protein synthesis leading to parasites in the quiescence state, and (iii) active drug efflux. The upregulated expression of cathepsin-L like protease, amastin-like surface protein, and amino acid transporter and downregulated expression of the gene encoding ABCG2, pteridine receptor, adenylatecyclase-type receptor, phosphoaceylglucosamine mutase, and certain hypothetical proteins are concordant with genomic alterations suggesting their potential role in drug resistance. The study provided an understanding of the molecular basis linked to artemisinin resistance in *Leishmania* parasites, which may be advantageous for safeguarding this drug for future use.

## 1. Introduction

Leishmaniasis is a major public health problem affecting the poor population of the world, mainly in the developing countries. The disease is endemic in 97 countries with 70,0000 to one million new cases per year [1,2]. Visceral leishmaniasis (VL), caused by the protozoan *Leishmania donovani*, is the most severe type, with frequent outbreaks and a greater mortality potential. In 2018, more than 95% of new cases reported to World Health Organization (WHO) occurred in 10 countries, including India [1,3]. Due to the lack of a vaccine and effective vector control, management of VL relies exclusively on a handful of chemotherapeutic agents, but most of the therapeutics, including pentavalent antimonials, miltefosine, and liposomal amphotericin B, are associated with serious drawbacks, such as being toxic and expensive, with a declining efficacy pertaining to an increase in the occurrence of resistance [4,5,6]. Therefore, there is a need to explore new safe, effective, and affordable treatment options for VL.

The antimalarial drug artemisinin and its derivatives have been found to also be effective against non-malarial parasites, such as *Leishmania*. There are several in vitro and in vivo studies demonstrating the antileishmanial activity of artemisinin and its derivatives with a high safety index [7,8,9]. As far as the mechanism of action is concerned, artemisinin and its derivatives have been reported to cause programmed cell death in *Leishmania* promastigotes by a loss of mitochondrial membrane potential, enabling externalization of phosphatidylserine, DNA fragmentation, and cell cycle arrest at the sub-G0/G1 phase [10]. The drug also works by the restoration of normal nitric oxide (NO) production by infected macrophages, initially impaired due to infection with *Leishmania* parasites [11,12,13]. Further, studies in mice suggest that administration of artemisinin results in the generation of iron-artemisinin adducts, which causes clearance of intracellular amastigotes [14]. However, antileishmanial activities and the possible mechanism of resistance to artemisinin in *Leishmania* parasites have been poorly explored. An understanding of the mechanisms of drug resistance in *Leishmania* is vital to protect existing drugs and for the development of new ones [15]. Drug-resistant parasites apply various strategies in order to survive under drug pressure, such as reduced drug uptake, active drug efflux, alteration of the drug targets, inactivation of drugs, etc. [16,17,18,19,20,21,22]. Various transcriptomic studies of drug-sensitive vs. -resistant parasites revealed that a number of genes have altered expression in drug-resistant parasites. Our group has previously shown by microarray analysis that approximately 3.9% and 2.9% of the total *Leishmania* genome representing various functional categories, such as metabolic pathways, transporters and cellular components among others, were differentially modulated (>2 fold) in experimentally selected miltefosine- and paromomycin-resistant lines, respectively [23,24].

Whole-genome sequence (WGS) analysis is another important tool used to detect mechanisms of drug resistance in *Leishmania*. It was earlier reported that in the absence of transcriptional control, *Leishmania* parasites have evolved mechanisms to alter mRNA levels by increased gene dosage through gene amplification, gene deletion, and aneuploidy in order to adapt to stress conditions, such as drug pressure [25,26,27,28]. The genome sequence of *Leishmania* field isolates from the Indian sub-continent revealed gene copy number variation (CNV) to be associated with susceptibility to sodium stibogluconate (SSG) [29]. Similarly, aneuploidy has been observed in the context of antimony, methotrexate, and nelfinavir resistance; however, the link between aneuploidy and drug resistance was circumstantial [25,26,30,31,32]. Additionally, single-nucleotide polymorphisms (SNPs) in drug targets or key enzymes constitute another strategy to survive under drug pressure. The acquisition of an inactivation mutation in the *L. donovani* miltefosine transporter gene (*Ld*MT) and/or its β-subunit (LdRos3) was reported to increase miltefosine resistance in both in vitro and in vivo studies as well as in clinical isolates [33,34,35,36,37,38].

Artemisinin resistance in malaria is associated with SNPs on chromosome 10, 13, and 14, and non-synonymous SNPs in the propeller domain of a kelch gene located on chromosome 13 [39,40]. Analysis of the transcriptome of *Plasmodium falciparum* isolates revealed a higher expression of unfolded protein response (UPR) in artemisinin resistance. Previously, we explored the mechanism of artesunate (a derivative of artemisinin) resistance in *Leishmania* parasites and showed that artesunate resistance in *Leishmania* is associated with parasite virulence, host immune modulation, and unfolded protein responses [41]. In the present study, the genome and transcriptome of artesunate-sensitive vs. -resistant *Leishmania* parasites were analyzed using next-generation sequencing (NGS) and single-color DNA microarray technology, respectively. Analysis of the genome structure and modulated gene expression identified several genes/pathways, which were further validated for their role in the selection of artesunate resistance in *Leishmania*. Expression analysis of Heat shock protein 70 (Hsp70) and Aquaglyceroporin 1 (AQP1) was validated in K133WT and K133AS-R cell lysate. In view of the important roles of ATP-binding cassette protein (ABC) transporters and the AQP1 gene in drug resistance in *Leishmania*, their roles were explored in artesunate resistance using respective inhibitors. Based on the analyses, a model was predicted for artesunate resistance in *Leishmania*.

## 2. Materials and Methods

### 2.1. Parasite and Culture Condition

*L. donovani* field isolate (K133WT), earlier derived from bone marrow aspirates of a VL patient and cryopreserved in a lab, was revived and propagated in medium M199 (Sigma-Aldrich, St. Louis, MO, USA) supplemented with 10% heat-inactivated fetal bovine serum (HI FBS, Gibco, Waltham, MA, USA), 100 IU/mL penicillin G, and 100 mg/mL streptomycin at 26 °C. The isolate was exposed to increasing concentrations (up to 50 μM) of artesunate drug (Sigma Aldrich, St. Louis, MO, USA) to obtain experimental artesunate-resistant parasites, which were designated as K133AS-R. The susceptibility of K133WT and K133AS-R parasites towards artesunate was determined, which showed that there was a 3.73-fold increase in the mean IC_50_ (50% inhibitory concentration) of K133AS-R parasites at the promastigote stage, with a value of 78.63 ± 9.17 μM vs. 21.08 ± 3.15 μM, and a >3-fold increase in the mean IC_50_ at the amastigote stage, with a value 73.09 ± 1.14 μM vs. 21.62 ± 3.24 μM for K133AS-R vs. K133WT isolates. This was reported in our previous study [41].

### 2.2. Genomic DNA Isolation from Parasite Culture

Genomic DNA (gDNA) from K133WT and K133AS-R promastigotes was isolated using a Wizard Genomic DNA purification kit (Promega, Madison, WI, USA) following the manufacturer’s instructions. Quantification of the DNA was performed by optical density measurements in a Nanodrop and QubitFlex^®^ 2.0 Fluorometer (Thermo Fisher Scientific, Waltham, MA, USA). The quality of the gDNA was checked on 1% agarose gel for the single intact band.

### 2.3. Genomic Library Preparation and Sequencing

Preparation of paired-end (PE) sequencing libraries of K133WT and K133AS-R was initiated with 200 ng of genomic DNA using a Truseq Nano DNA Library preparation kit (Illumina, Inc., SanDiego, CA, USA). The generated library was examined in a Bioanalyzer 2100 (Agilent Technologies, Santa Clara, CA, USA) using a high-sensitivity (HS) DNA chip and sequenced using the Illumina Hiseq 2000 platform according to the manufacturer’s standard cluster generation and sequencing protocol [42]. Briefly, the mechanical shearing of gDNA by a Covaris instrument (Woburn, MA, USA) was done to generate fragments of 250–350 bp, after which fragmented ends were repaired and tailed with A at 3′. Thereafter, adapters were ligated, which was necessary for binding dual-barcoded libraries to the flow cell for sequencing. Finally, 314–355-bp libraries were generated and high-fidelity PCR amplification was done using HiFi PCR master reaction component mix to ensure maximum yield from limited amounts of starting material for sequencing on an Illumina Hiseq 2000 platform (Illumina Inc., San Diego, CA, USA) (2 × 150 bp chemistry). Whole-genome sequencing resulted in the generation of approximately 4 GB data per sample. The sequences of *L. donovani* K133WT and K133AS-R are available with NCBI GenBank as a BioProject with SRA accession no. PRJNA657979.

### 2.4. Whole-Genome Sequencing Data Analysis

Genomic data analysis was executed with minor modifications as described previously by Dumetz et al. 2017 [43]. The paired-end (PE) raw reads obtained from the sequencer were checked for the quality of the reads using FastQCv0.11.8 (Babraham Institute, Cambridge, UK) and were further trimmed to improve the quality of the reads using the Trimmomatic tool v0.38 (Usadellab. org, RWTH Aachen University, Germany) [44]. The *L. donovani* strain LdBPK282A1 reference genome was indexed and high-quality pair-end reads were mapped using Burrows-Wheeler Aligner (BWA-MEM v0.7.5a algorithm, Broad Institute, Cambridge, MA, USA) [45]. The generated SAM file was converted into BAM format and duplicates were removed using Picard toolkit v1.119 (Broad Institute, Cambridge, MA, USA). Further, the BAM file was used for identifying SNPs and InDels using GATK Haplotype caller v3.4 (Broad Institute, Cambridge, MA, USA). Filtering of SNPs and InDels was performed using Bcf tools v0.1.18 (Sanger Institute, Cambridge shire, UK) from subdirectory of SAM tools (mapping quality cut off 25 and read depth of 15) and the variants were annotated with the SnpEff v4.3 tool (McGill University, Montreal, QC, Canada) [46,47].

Estimation of CNV along with chromosomal somy was done in accordance with the protocol designed by Downing et al. 2011 [29]. CNV estimation was done using CNVnator (https://github.com/abyzovlab/CNVnator) [48], and for somy assessment, the median read depth of each chromosome (*d_i_*) was computed first followed by median depth estimation of 36 complete chromosomes (*d_m_*). The somy state of an individual chromosome is determined as the ratio of (*d_i_/d_m_*) and the chromosome ploidy value is specified as 2 × (*d_i_/d_m_*), considered often for diploid species [38]. The full cell-normalized chromosome somy (S)-value: S < 1.5, 1.5 < S < 2.5, and 2.5 < S < 3.5, was assigned to monosomy, disomy, and trisomy, respectively [43].

### 2.5. Functional Annotation and Classification of Unigenes

To identify all the unigenes present in K133WT and K133AS-R, a homology search was performed against the NCBI non redundant (NR) protein database in accordance with BLASTx program (NCBI, Bethesda, MD, USA) using a cutoff E-value of 10^−05^ and the maximal aligned results with the lowest E-value were chosen to annotate the unigenes [49,50]. The Gene Ontology (GO)-based annotation of the unigenes was carried out using Blast2GO version 3.0 (Biobam, Valencia, Spain) and Web Gene Ontology Annotation Plot (WEGO) was utilized to designate GO classification on the basis of the distribution of gene functions in different species [51,52,53,54]. The basis of the functional classification considered was biological processes, cellular components, and molecular functions.

### 2.6. Total RNA Isolation from Parasites

Early log-phase promastigotes (1 × 10^8^) of both K133WT and K133AS-R were used to isolate total RNA using TRIzol reagent according to the manufacturer’s instruction. Extracted RNA was cleaned up using a RNeasy Plus mini kit (Qiagen, Hilden, Germany). The absorbance of purified RNA was taken at 260 and 280 nm using a Nanodrop Spectrophotometer (Thermo Fisher Scientific, Waltham, MA, USA). The quality and integrity of RNA were assessed on an RNA 6000 Nano Assay Chips on Bioanalyzer 2100 (Agilent Technologies, Santa Clara, CA, USA). RNA of good quality based on the 260/280 values (Nanodrop, Thermo Scientific, Waltham, MA, USA), rRNA 28S/18S ratios, and RNA integrity number (RIN) was used for further analysis [24].

### 2.7. Oligonucleotide Array

Global mRNA expression profiling of K133WT and K133AS-R *L. donovani* was carried out usingsingle color microarray-based gene expression profiling. A high-density *Leishmania* multispecies 60-mer oligonucleotide array slide [8 × 15 K format] was used for the microarray experiment. The slide represented the entire genome of *L. infantum* and *L. major*. The microarray chip printed by Agilent Technologies (Santa Clara, CA, USA), contained a total of 9233 *Leishmania*-specific genes, including 540 control probes as described earlier [24,55,56].

### 2.8. RNA Labelling, Amplification, Hybridization, and Data Analysis

First, 200 ng of total RNA were converted to cDNA using oligodT primer tagged to T7 polymerase promoter at 40 °C. cDNA thus obtained was converted to cRNA using T7 RNA polymerase enzyme. The dye Cy3 was also incorporated during this step. Labeled cRNA was then cleaned using Qiagen RNeasy Mini kit columns (Qiagen, Cat No: 74106, Hilden, Germany) and quality assessment was carried out using the Nanodrop ND-1000. Following this, Cy3-labeled cRNA was fragmented at 60 °C. Fragmented cRNA was hybridized on the array (AMADID: 027511) using the Gene Expression Hybridization kit (Agilent Technologies, Santa Clara, CA, USA) at 65 °C for 16 h in Sure hybridization Chambers. Hybridized slides were washed using Agilent Gene Expression wash buffers (Agilent Technologies, Santa Clara, CA, USA) and scanned on an Agilent Microarray Scanner (Agilent Technologies, Part Number G2600D). Images thus obtained were quantified using Agilent’s Feature Extraction Software Version-10.7 (Santa Clara, CA, USA). Feature-extracted raw data were analyzed using the GeneSpring GX12.6.1 microarray data and pathway analysis tool (Santa Clara, CA, USA). Quartile (75th percentile) normalization was performed. Storey and bootstrapping analysis was performed for multiple testing corrections. The expression profile of K133AS-R parasites was extrapolated on a chromosome map of *Leishmania* parasites using custom R programs. The modulated expression of genes was identified using two criteria: (a) statistical and (b) biological. Statistical significance was determined by the t-test (unpaired) and a *p* value < 0.05 was considered as significant for both K133WT and K133AS-R parasites. The biological cutoff for up- or downregulation was 2-fold. Differentially regulated genes were analyzed for functional classification using the GeneDB, BLAST2GO, and AmiGO databases. The pathway analysis was carried out using the gene Spring GX12.6.7 (Santa Clara, CA, USA) and KEGG pathway analysis tool (Bethesda, MD, USA). Interacting partners of up- or downregulated genes in K133AS-R parasites were identified using the String 9.01 database [24].

### 2.9. Data Availability

The complete genome sequence was deposited in GenBank as BioProject number PRJNA657979: for *L. donovani* K133AS-R under the SRA accession number SRR12487478 and BioSample number SAMN15854505 and for *L. donovani* K133WT under the SRA accession number SRR12487479 and BioSample number SAMN15854504. The microarray data were deposited in the GEO NCBI database (http://www.ncbi.nlm.nih.gov/geo) in the MIAME format (GEO accession number GSE118460).

### 2.10. Quantitative Real-Time PCR (qPCR)

A total of 14 genes were selected from microarray data and validated for their differentially modulated expression by q-PCR (Appendix A). First-strand cDNA was synthesized, from 5 μG of total RNA isolated from K133WT and K133AS-R promastigotes (early log phase), using the Superscript II RNAse H reverse transcriptase enzyme (Invitrogen, Carlsbad, CA, USA) and OligodT primers (Fermentas, Waltham, MA, USA). Equal amounts of cDNA were amplified in 25-μL reactions (in triplicate) containing 6 pmoL forward and reverse primers and 1 X Fast SYBR Green mastermix using a ABI 7500 Real-time PCR system (Applied Biosystems, Waltham, MA, USA). The relative amount of PCR products generated from each primer set was determined based on the threshold cycle (Ct) value and the amplification efficiencies. Gene expression levels were normalized using constitutively expressed genes encoding cystathionine-β-synthase (CBS) and glyceraldehyde-3-phosphate dehydrogenase (GAPDH). Quantification of the relative changes in the target gene expression was calculated using the 2^−ΔΔCt^ method. Primers for the targeted genes were designed using Primer express software version 3.0 (Applied Biosystems, Waltham, MA, USA) [57]. The list of genes, their functional relevance, and the primers used for real-time PCR are given in Appendix A.

### 2.11. Western Blotting of Promastigote Cell Lysate

Preparation of the parasite lysate and Western blot analysis was performed following the method described earlier [58]. K133WT and K133AS-R cell lysates (100 µG) were separated by sodium dodecyl sulphate polyacrylamide gel electrophoresis (SDS-PAGE) on a 12% polyacrylamide gel and transferred to nitrocellulose membranes. The membrane strips were blocked and incubated sequentially with anti-AQP1 (1:1000), anti-HSP70 (1:500), or anti-tubulin (1:1000) (endogenous control) primary antibodies. Following this, the membrane was probed with Horse radish Peroxidase (HRP)-conjugated anti-rabbit IgG (1:80,000) produced in mice (Sigma Aldrich, St. Louis, MO, USA). Blot was developed using Western blot detection enhanced chemiluminescence (ECL) detection reagent (Merck, Burlington, MA, USA). The image was scanned with ChemiDoc (Bio-Rad, Hercules, CA, USA) and analyzed using Image Lab™ 5.1 software (Bio-Rad, Hercules, CA, USA) [58].

### 2.12. Cytotoxicity Assay

The 3-(4,5-dimethylthiazol-2-yl)-2,5-diphenyltetrazolium bromide (MTT) assay was used to assess the cytotoxicity of the inhibitors towards host macrophages and was performed following the manufacturer’s instructions. Primary peritoneal macrophages (PECs) extracted from Balb/c mice were incubated with AQP1 inhibitor (Tocris Biosciences, Bristol, UK) (12.5–400 µM) or ABC transporters modulator verapamil (Sigma Aldrich, St. Louis, MO, USA) (6.25–200 µM) in a 96-well plate for 48 h at 37 °C in 5% CO_2_. Following this, 25 μL of (5 mg/mL in 1 × PBS) MTT were added to each well and the plate was re-incubated at 37 °C in the dark. Then, 4 h later, all media was removed and 150 μL of Dimethyl Sulfoxide (DMSO) were added to each well, mixed well by pipetting, and incubated for 15 min in the dark. Absorbance was taken at 540 nm on an Infinite M200 multimode reader (Tecan, Switzerland). A decrease in the absorbance at 540 nm indicated a decrease in cell viability.

### 2.13. Artesunate Susceptibility in the Presence of Inhibitors

The susceptibility of K133WT and K133AS-R parasites towards artesunate was determined in the presence of the AQP1 inhibitor (Tocris Biosciences, Bristol, UK) and ABC transporter modulator, verapamil (Sigma, St. Louis, MO, USA). At the promastigote stage, both K133WT and K133AS-R isolates (1 × 10^5^) were seeded into a 96-well plate with various concentrations of artesunate drug (1–650 µM) alone or in the presence of 40 µM of AQP1 inhibitor or 8 µM of verapamil and incubated at 25 °C. After 72 h of incubation, 50 μL of Resazurin (Sigma Aldrich, St. Louis, MO, USA) (0.0125% (*w*/*v*) in Phosphate Buffered Saline (PBS) were added to each well and the plates were further incubated for 18 h. Fluorescence was measured at an excitation wavelength of 550 nm and emission wavelength of 590 nm on an Infinite M200 multimode reader (Tecan, Switzerland) to determine cell viability. Sigmoidal regression analysis was used to calculate IC_50_ [24].

At the amastigote stage, the mice PECs were infected with late log-phase promastigotes of K133WT or K133AS-R at a ratio of 10 parasites: 1 macrophage, plated into 8-well chamber slides and incubated for 16 h at 37 °C in 5% CO_2_. Non-internalized promastigotes were washed off and infected macrophages were further incubated with various dilutions of artesunate drug (13, 26, 52, 104, 208, and 260 μM) with or without AQP1 inhibitor (Tocris Biosciences, Bristol, UK) (40 µM) or verapamil (8 µM). The inhibitor/modulator alone at the tried concentration was not lethal to either K133WT/K133AS-R isolates or host macrophages. Then, 48 h later, the slides were fixed and stained using Diff-Quik solutions. Macrophages were then examined for intracellular amastigotes at 1000 × magnification. The number of *L. donovani* amastigotes per 100 macrophages was counted and the survival rate of parasites relative to untreated macrophages was calculated to determine the IC_50_ value [24].

### 2.14. Ethics Approval

The ethics approval was obtained from the Institute Animal Ethics Committee of the ICMR-National Institute of Pathology, Safdarjung Hospital campus, New Delhi, India (Project No. NIP/IAEC-1502). The procedures for the care, use, and euthanasia of experimental animals were carried out under the guidelines of the Committee for the Purpose of Control and Supervision of Experiments on Animals (CPCSEA, Indira Paryavaran Bhawan, Jor Bagh, New Delhi) Government of India.

## 3. Results

### 3.1. Whole-Genome Sequence Diversity Data of K133AS-R Compared to K133WT

Comparative WGS data analysis of both in vitro-generated artemisinin-resistant parasite (K133AS-R) and the wild-type field isolate (K133WT) was performed to decipher the mechanisms responsible for drug resistance. Detailed analysis of SNPs and insertion-deletion mutations (InDels) was performed for K133WT and K133AS-R isolates relative to the *L. donovani* reference using GATK’s Haplotype Caller (HC) [59]. WGS data analysis of K133WT showed a higher number of upstream gene variants followed by intergenic region and missense gene variants. Out of a total of 341 gene variants, 191 SNPs and 150 InDels were observed (Figure 1A). The maximum number of SNPs was observed on chromosome number 34, while no SNP was observed on chromosome number 5, 9, 11, 14, 21, and 26 out of a total of 36 chromosomes in *Leishmania*. Amongst the total InDels, 114 nucleotide insertions and 36 nucleotide deletions were observed.

The artemisinin-resistant parasite generated in vitro under drug selection pressure (K133AS-R) showed a higher number of upstream gene variants followed by intergenic region gene variants. Out of a total of 477 gene variants, 240 SNPs and 237 InDels were observed (Figure 1A).

The maximum number of SNPs was observed on chromosome number 31, while no SNPs were found on chromosome number 14 and 26, which is a common observation among K133WT and K133AS-R (Figure 1B). Among InDels, 173 nucleotide insertions and 64 nucleotide deletions were observed. Unique gene variants were also observed among K133WT and K133AS-R. In K133WT, the unique gene variants identified were upstream variant 254 (74.48%), downstream 23 (6.74%), missense 27 (7.9%), frameshift 7 (2.05%), and intergenic region gene variants 27 (7.9%), while other variations included disruptive / conservative in-frame insertions, one each and one conservative in-frame deletion, which is 0.3% of the total gene variation observed. In case of K133AS-R, the unique gene variants observed were 357 (74.84%) in upstream, 45 (9.43%) in downstream, 47 (9.85%) in intergenic region, 16 (3.35%) missense variant, 11 (2.05%) frameshift variants, and one stop-lost splice variant (0.21%) (Figure 1C). Insertions were observed to be the highest in the genome followed by transition, transversion, and deletion.

### 3.2. Differentially Expressed Genes in K133AS-R vs. K133WT

The data for K133AS-R revealed several differentially expressed genes, which are expected to contribute to drug resistance. Extensive variation in the expression of several genes like pteridine transporter and histone-encoding genes was observed in artemisinin-resistant isolates. Marked variation in the number of peptidases, such as metallopeptidase (LDBPK_330210), aminopeptidase P1 (LDBPK_020010), and lipases (LDBPK_341140), was also observed in K133AS-R. Enzymes involved in the lipid biochemical pathway, such as fatty acid elongation and fatty acid desaturation, were affected in artemisinin-resistant *Leishmania*, suggesting a decreased fluidity of the parasite membrane, which may be contributing towards drug resistance as observed in the case of miltefosine-resistant parasite [60]. Genes encoding phosphoglycan β-1,3 galactosyltransferase (involved in glycosylation of proteins), ATP binding cassette transporters (ABC transporters), ABCA2, ABCA7, and ABCA8 exhibited one missense and two frameshift mutations having moderate and high impact in K133AS-R parasites. Additionally, changes in folate/biopterin transporter (upstream gene variant, impact modifier), P-type H^+^-ATPase (frameshift mutation), and UDP-galactose transporter have been observed in the AS-R parasite. Cell surface protein-encoding genes viz. amastin-like proteins, and proteophosphoglycan (ppg3)-related protein displayed mutation in the K133AS-R isolate. Alterations in ceroidlipofuscinosis neuronal protein 3 (CLN3, LDBPK_061360) responsible for *Leishmania* virulence were also observed, showing six mutations, including missense mutation with moderate impact, which may have a direct effect on lysosomal function [61]. Moderate impacts on enzymes of the TCA cycle viz. citrate synthase, pyruvate kinase, and succinate dehydrogenase were noted in K133AS-R. In addition, specific genes present in K133AS-R that encode peptidase-like cysteine peptidase B, serine peptidase, heat shock proteins, upstream gene variants, and downstream gene variant with modifier impact are speculated to have direct or indirect role in pathogenesis. Interestingly, two novel gene mutations in K133AS-R including apical membrane antigen1 (AMA1, LDBPK_301480) (moderate impact, missense mutation) and cathepsin L-like protease were identified, whose role in pathogenesis has been reported previously [62].

### 3.3. Chromosomal Diversity in Artesunate-Resistant L. donovani

#### 3.3.1. Chromosome Copy Number Variation (CNV)

Chromosome copy number analysis revealed large differences between K133WT and K133AS-R. The gene copy number variants’ length observed was from 0.2 to 200 kilo base pair (kb) based on the size distribution of identified CNVs (Figure 2A). Of the total CNVs observed in K133AS-R, most were in the range of 1–5 kb size followed by 20 to 100 kb, accounting for 43.66% and 16.39%, respectively. A comparative analysis of local gene copy number variations between K133WT and K133AS-R was performed. In case of K133WT, a total of 586 CNVs were identified in which 365 deletions and 221 duplications were observed compared to K133AS-R, in which a total 616 CNVs were identified out of which 377 were deletions and 239 were duplications (Figure 2B).

#### 3.3.2. Variance in Allelic Frequency Due to Change in Chromosomal Somy/Ploidy

In most of the cases, drug resistance in pathogenic microorganism correlates with gene expression changes, which somehow are concordant with the chromosomal ploidy changes. Normalized read depth data are generally used to assess copy number variation as somy estimation does not always show a result in integral values, since it depicts the average of a population of the cell that does not strictly show identical karyotypes. To determine somy, the two-loop method was used [38,63]. Chromosomal somy data analysis of K133WT and K133AS-R shows that most of the chromosomes were disomic. Deflection from this pattern was detected in chromosome number 14 and 32, which were monosomic, while chromosome 5, 8, 20, 23, and 31 displayed the trisomy condition in both K133WT and K133AS-R. Chromosome 12 only was found in the trisomy condition, a unique observation in K133AS-R (Figure 3).

### 3.4. Functional Annotation and Classification of K133WT and K133AS-R Unigenes

Additional validation, functional annotation, and classification of K133WT and K133AS-R unigenes derived from reference-based assembly data was performed as described in the methodology section. Out of 7671 genes retrieved in K133WT and 7792 genes in K133AS-R, a total of 7652 (99.75%) in K133WT and 7778 (99.82%) genes were found with BLAST hits. K133AS-R unigenes were annotated with at least one biological term from GO information, while the remaining 19 genes in K133WT and 14 in K133AS-R did not result in any BLAST hit. Species distribution analysis based on BLASTx results with BLAST hit sharing showed high sequence similarity with *L. donovani* and *L. infantum* sequences (Figure 4A,B). In K133WT, 2925 GO terms were allocated to biological processes, 2987 terms to cellular components, and 3152 GO terms to molecular functions, while in K133AS-R, 2942 GO terms were assigned to biological processes, 3005 terms to cellular components, and 3178 GO terms to molecular functions. Within the biological process category, cellular metabolism, cellular component organization, or biogenesis was most abundant. Within the cellular component category, GO terms corresponded to the cell and organelle part, membrane part, and protein-containing complexes (Figure 4C,D). Under molecular function category, GO terms mostly corresponded to different catalytic, binding, and transporter activity, which were abundant among unigenes [52].

### 3.5. Comparative Transcriptome Analysis of K133WT vs. K133AS-R Parasites

Gene expression analysis using one-color DNA microarray experiment, of K133WT vs. K133AS-R isolate, revealed a modulated expression of 208 genes (approximately 2.26%) in drug-resistant parasites. The plot log_2_ transformed expression ratio of K133AS-R (red line) vs. K133WT (green line) as a function of the chromosomal location of microarray probes is shown in Appendix A. Out of 208 differentially modulated genes, 102 genes (1.11%) were upregulated and 106 genes (1.15%) were downregulated in K133AS-R parasites. The overall expression pattern of mRNA is shown in Appendix A.

The gene expression level on the genomic scale was analyzed using a chromosome map (Figure 5A). The chromosome map showed that chromosome 18, 25, 31, and 33 contained higher numbers of upregulated genes while chromosome 33 and 36 contained higher numbers of downregulated genes in the K133AS-R isolates. Among the upregulated genes, the highest number were present on chromosome 31, which included AQP1 (LinJ.31.0030), amastin (LinJ.31.0460, LmjF.31.0450), and a few uncharacterized proteins. Upregulated proteins include autophagocytosis protein (LinJ.33.0320), protein having RNA ligase (LinJ.33.0580) activity and transaminase (LinJ.33.1410) activity on chromosome 33, protein involved in trpanothione biosynthesis process (LinJ.18.1660) on chromosome 18, Kinesin (LinJ.25.2150), and DNA-directed RNA polymerase II (LinJ.25.1350) on chromosome 25. The maximum number of downregulated genes were present on chromosome 33, among which more than 50% were hypothetical uncharacterized proteins. Other downregulated genes on chromosome 33 included translation initiation factor 2 (LinJ.33.2880), small nuclear ribonucleoprotein complex (LinJ.33.3340), H1 histone-like protein (LinJ.33.339/0), and metallocarboxipeptidase (LinJ.33.2670). Genes showing downregulated expression on chromosome 36 included isoleucyl-t-RNA synthetase (LinJ.36.5870), translation elongation factor 1-β (LinJ.36.1490), glucose transporters (LinJ.36.6550, LmjF.36.6290, LinJ.36.6560), phosphoglycerate mutase family member 5 (LinJ.36.4270), and ubiquitin protein ligase (LinJ.36.6600).

Genes showing differential expression (both up- and downregulated genes) were classified into various functional categories and a number of altered pathways in K133AS-R parasites were identified with the help of several databases and bioinformatics tools as mentioned above in Section 2.8. The percentage of genes exhibiting altered expression with genes remained unaltered in K133AS-R parasites is shown in Figure 5B. Among the 208 genes showing modulated expression in K133AS-R parasites, a total of 144 genes were categorized into function and distributed into eight different functional categories (Figure 5C). All the 144 genes with their functional categories are enlisted in Appendix A.

### 3.6. Validation of Modulated Gene Expression Using qPCR

Fourteen differentially expressed genes were selected for validation of expression analysis based on their role in various metabolic pathways and artesunate resistance. The selected 14 genes (9 upregulated and 5 downregulated) were validated for their expression in K133WT and K133AS-R parasites by qPCR. The fold change in the gene expression of K133AS-R/K133WT observed in q-PCR was compared with that observed in microarray experiments (Figure 6). The results obtained by qPCR for selected genes agreed with the transcriptome data derived by microarray experiments.

### 3.7. Targeted Protein Profiling of AQP1 and HSP70 in K133WT and K133AS-R Leishmania Parasites by Western Blotting

Western blot analysis revealed that the expression of AQP1 was 1.6-fold higher in K133AS-R parasites whereas that of HSP70 was 5.46-fold lower in K133AS-R parasites as compared to K133WT parasites (Figure 7).

### 3.8. Susceptibility of K133WT and K133AS-R Parasites in the Presence of the AQP1 Inhibitor and Modulator of ABC Transporters

The susceptibility of K133WT and K133AS-R parasites towards artesunate was determined in the presence of AQP1 inhibitor and modulator of ABC transporters, verapamil. The cytotoxicity of the AQP1 inhibitor or verapamil determined for host macrophages (mice PECs) by the MTT assay revealed that the cytotoxic concentration 50% (CC_50_) of the AQP1 inhibitor was 233.47 ± 40.19 and that of verapamil was 111 ± 14.17 (Appendix A). The IC_50_ of K133AS-R parasites towards artesunate significantly decreased by 1.9-fold in the presence of the AQP1 inhibitor and 2.2-fold in the presence of verapamil at the promastigote stage (Figure 8A). Surprisingly, at the intracellular amastigote stage, K133WT parasites showed a significant increase of >4-fold in the IC_50;_ however, no significant alteration was observed in the IC_50_ of K133AS-R parasites towards artesunate in the presence of the AQP1 inhibitor. Further, in the presence of verapamil at the amastigote stage, IC_50b_ of artesunate for the K133AS-R parasites decreased by 2-fold (Figure 8B). However, there was no significant alteration in IC_50_ of K133WT parasites in the presence of the AQP1 inhibitor at the promastigote stage and in the presence of verapamil at the promastigote or amastigote stage (Figure 8A,B).

### 3.9. Analysis of Modulated Genes and Pathways in K133AS-R Parasites

Based on all the observations, a model depicting all the adaptations in K133AS-R parasites was proposed (Figure 9), which suggests the following genes/pathways are affected in K133AS-R parasites.

#### 3.9.1. Autophagy, UPR, and Oxidative Stress

In K133AS-R, Atg8 (LinJ.19.0860) that plays an important role in formation of autophagosome was downregulated. Downregulated expression of HSP70 was also observed in K133AS-R parasites. On the other hand, an upregulated expression of lipoate protein ligase (LinJ.36.3230) involved in lipoic acid biosynthesis was observed in K133AS-R parasites, which eventually lead to upregulated expression of γ-glutamyl cysteine synthetase (GSH1).

#### 3.9.2. Carbohydrate, Lipid, and Amino Acid Metabolism

K133AS-R parasites showed downregulated expression of gene phosphoacetylglucosamine mutase (LmjF07.0805) involved in the conversion of N-acetyl-α-D-glucosamine-1-phosphate to N-acetyl-D-glucosamine-6-phosphate, which later forms fructose-6-phosphate. Further, various glucose transporters, such as the glucose transporter, Imgt2 (LinJ.36.6550, LmjF36.6290), were also downregulated, suggesting downregulation in carbohydrate metabolism. On the other hand, genes involved in amino acid and lipid metabolism, such as methylmalonyl CoA mutase (LinJ.27.0310; involved in isoleucine, valine, and leucine metabolism), glutamine aminotransferases (LinJ.33.1410; involved in glutamine metabolism), myo-inositol-1-phosphate synthase (LinJ.14.1450), and a hypothetical protein having lipase activity (LinJ.13.0200) involved in lipid metabolism, showed upregulated expression.

#### 3.9.3. DNA Synthesis and Translation Machinery

Genes responsible for DNA replication like nucleoside transporter 1 (LinJ.36.2040), H1 histone-like protein (LinJ.33.3390), and endonuclease/exonuclease activity (LinJ.28.1000), were downregulated. Genes involved in protein translation, such as translation initiation factor IF-2 (LinJ.33.2880), Isoleucyl-tRNA synthetase (LinJ.36.5870), and 28S ribosomal RNA (LmjF27.rRNA.32), were also downregulated. On the other hand, small RNA molecules that play an essential role in RNA biogenesis and guide chemical modifications of ribosomal RNAs (rRNAs) and other RNA genes (tRNA and snRNAs) U1snRNA, U2 snRNA, and U3 snRNA were upregulated. Further, genes involved in protein degradation, such as metallopeptidase (LinJ.11.0640) and carboxypeptidases (LinJ.33.2670), were downregulated.

#### 3.9.4. Modulated Expression of Transporters

Aquaglyceroporin (AQP1) (LinJ.31.0030), UDP-galactose transporter (LPG5B) (LinJ.18.0400), and ATP-binding cassette protein subfamily G, member 1, putative (ABCG1) (LmjF06.0080) were upregulated in K133 AS-R parasites. On the other hand, an ABC transporter family-like protein (LinJ.33.3410A), nuclear transport factor 2 (LinJ.10.0900), and amino acid transporters, AAT19 (LinJ.07.1340) and AAT22 (LinJ.22.0100), showed downregulated expression.

### 3.10. Correlation of Whole-Genome Sequencing Analysis with Transcriptomic Data

Advancement in genomics and transcriptomics technologies have conferred considerable enhancement of our knowledge related to the set of changes that occur within the parasite at the molecular level resulting in the evolution of drug resistance. Comparative analysis of the genome and transcriptome data of the two distinct strains of *L. donovani* (K133WT and K133AS-R) provided a correlation of the mechanism behind the development of artemisinin resistance. Eight upregulated and 10 downregulated genes in the transcriptome data matched with the NGS data. Out of eight upregulated genes, two amastin-like surface protein genes (LinJ08_V3.0700/LdBPK_080710) and (LinJ34_V3.0700/LdBPK_43111505) showed one and five mutations respectively; the remaining six genes had upstream gene variants. The types of mutations observed were insertion, deletion, and transition. Genes that correlated in both included the amino acid transporter ATP11 and cathepsin—L like cysteine protease. Out of 10 downregulated genes, ABCA2 (LinJ11_V3.1230/LdBPK_111210) and receptor-type adenylatecyclase b (fragment) (LinJ17_V3.0140/LdBPK_170120) displayed a frameshift mutation at two sites. The genes that were observed to be downregulated included phosphoacetylglucosamine mutase-like protein (LinJ07_V30930/LdBPK_070930) and pteridine transporter (LinJ06_V3.1320/LdBPK_061320). Cathepsin L-like protease, a type of lysosomal endopeptidases, is present in both the promastigote and amastimogote stage of *Leishmania* species and involved in crucial biological process of parasites, such as evasion of the host immune system [63,64,65].

## 4. Discussion

Sesquiterpene, artemisinin, a secondary metabolite extracted from *Artemisia annua*, is an important antimalarial drug that has shown antimicrobial and antiviral activities [66,67]. Several in vitro and in vivo studies suggested potential antileishmanial activity of this drug [8,9,10,68]. However, the possibility of the emergence of resistance following the use of artemisinin as antileishmanial treatment cannot be denied. In our previous study, we reported that in vitro-selected artesunate-resistant *Leishmania* parasites were more virulent, successfully modulating the host cell defense mechanism, and exhibited altered expression of genes involved in the unfolded protein response, as compared to sensitive parasites [41]. The present study aimed to explore the genome and transcriptome of artesunate-resistant *Leishmania* parasites in order to understand the mechanism of resistance and to safeguard this drug for future use. Next-generation sequencing (NGS) platforms have advanced to provide a precise and comprehensive means for the detection of molecular mutations. Genomic and transcriptomic analyses would help in the advancement of our understanding of the biology of *Leishmania*. This comparative analysis of whole-genome sequences attempted to explicate genetic factors responsible for drug resistance in *L. donovani*. Here, we demonstrated that the in vitro-selected artesunate-resistant (K133AS-R) parasite was quite distinct from the sensitive wild-type (K133WT) at the genome and transcriptome level.

Major findings of the study are summarized in three sections. Firstly, from the genomic landscape, we found a high number of SNPs and InDel, many of them having a pronounced influence (stop codon gained/lost and frame-shifts) on essential biological functions. Briefly, in K133AS-R, upstream gene variants were higher followed by intergenic region gene variants. Out of the total number of gene mutations, SNPs were high as compared to InDel. The highest number of SNPs was observed on chromosome number 12, 31, 34, and 35. Among InDel mutations, insertions were greater than deletions. Non-coding mutations, such as upstream gene variants and downstream gene variants, affect regulatory elements and lead a to loss of function that results in reduced gene expression, or a gain of function resulting in differential gene expression [69]. In this study, we analyzed that selective forces are majorly acting on non-coding regions of the genome. Secondly, the major changes observed were concerned with local copy number variations (CNVs). In K133AS-R, higher deletion occurred as compared to duplication and CNV lengths in the range of 1–5 kbp were either deleted or duplicated, depicting that changes occurred at small sequences rather than larger sequences. The highest number of CNVs were observed in K133AS-R on chromosome no 31, 29, 20, and 18. In the absence of regulation of gene expression at the initiation site, duplication/deletion of specific genes in a genomic sequence modulates the transcript level and its products [69,70]. Complex chromosomal copy number variation is often observed in *Leishmania* parasites due to their asexual mode of replication [26]. Thirdly, second-generation sequencing data obtained with the Illumina analyzer sets out remarkable read depth coverage throughout the chromosomes of *Leishmania*, and both the K133AS-R and K133 WT exhibited a uniform read depth in all chromosomes that is disomic except chromosome number 12 in K133 AS-R. A read depth greater than two-fold was observed in case of chromosome 12, suggesting that the chromosome is present in the trisomy condition. Chromosomy variation in *Leishmania* is a well-known adaptive strategy in response to experimental drug resistance selection [71]. Aneuploidy is mostly influenced by the environmental condition and is more prevalent in promastigotes under in vitro conditions than in amastigotes present inside the vertebrate host. It arises through unlicensed replication due to a lack of proper cell cycle regulation and/or mitotic non-disjunction [43]. Further, GO terms based functional annotation of genes lead to classification into different categories, including metabolic, cellular processes, and biological regulation, which include the response to stimulus, cell signaling, and growth. WGS data analysis gives ample information regarding genetic variation compared to other sequencing approaches that include SNPs, InDel, as well as structural variants *viz* CNVs, inversion, translocation, and ploidy variation in chromosomes [72].

Analysis of transcriptome data by microarray and further experimental validation of differentially expressed proteins resulted in several important findings. To maintain cellular homeostasis, the eukaryotic cells have developed specialized mechanisms, such as lysis of intracellular proteins and organelles, which regulate cellular functions like enzymatic activity, removal of toxic or misfolded proteins, and the production of free amino acids to ensure cell survival under stressful conditions. The eukaryotic cells are known to perform these functions by the process of autophagy, which is believed to have originated at a later point during evolution [73]. Artesunate causes high levels of ROS generation within the cell. Further, it has been reported in cancer cells that autophagy plays a cytoprotective role within cells by inhibiting ROS. In K133AS-R, downregulated expression of Atg8 suggested reduced inhibition of ROS. In addition, deceased expression of HSP70, both at transcript and protein levels, suggested an accumulation of a higher number of misfolded proteins, resulting in higher ER stress and finally higher ROS production in K133AS-R parasites. On the other hand, upregulated expression of lipoate protein ligase leads to upregulated expression of GSH and thus plays an important role in glutathione biosynthesis and response to oxidative stress. This may be a compensatory approach of K133AS-R parasites to survive under oxidative stress.

Downregulated expression of gene phosphoacetylglucosamine mutase, which is eventually involved in the formation of fructose-6-phosphate, an important component of glycolysis/gluconeogenesis, suggested downregulation of these pathways in K133AS-R parasites. In addition, the downregulated expression of various glucose transporters suggested that K133AS-R parasites may not depend on carbohydrate metabolism for energy requirements. Hence, artesunate-resistant parasites may depend on amino acids and lipids for energy generation as inferred by the upregulated expression of methylmalonyl CoA mutase (involved in isoleucine, valine, and leucine metabolism) and glutamine aminotransferases (involved in glutamine metabolism) while myo-inositol-1-phosphate synthase and a hypothetical protein are involved in lipid metabolism.

Leucin-rich AMA1 protein secreted by many *Leishmania* species, including *L. donovani*, helps them to interact with cholesterol present in the host cell membrane and thereby assist the internalization of parasites [74,75,76]. Amastin, a transmembraneglycoprotein, encoded by a large gene family initially reported in the amastigote stage of trypanosomes and later observed as a surface protein expressed in *Leishmania* species (encoded by six copies of genes) plays an important role in visceralization [77]. The importance of amastin in the pathogenesis of *Leishmania* species is well documented in a previous study [55]. The data analysis showed that the parasites may undergo genomic alterations to express certain genes differentially to adapt to the drug-induced selection pressure.

K133AS-R parasites showed downregulated expression of several genes involved in DNA synthesis and translation machinery. Reduced DNA/protein synthesis leads to an arrest of parasites in a quiescent state, which may be responsible for drug resistance as reported in case of artemisinin resistance in malaria [78]. Further, there was a downregulated expression of metallo- and carboxy-peptidase involved in protein degradation, which may be an adaptive approach of K133AS-R parasites to overcome reduced protein synthesis.

In *Leishmania*, AQP1 plays an important role in providing nutrients from the host organism, mainly glucose, amino acids, and fatty acids. These may also be responsible for discarding waste and metabolic end-products, such as lactate, from the parasite’s cytosol [79]. In the presence of AQP1 inhibitor, drug-resistant mutants showed a significant increase in susceptibility towards artesunate at the promastigote stage; however, no significant alteration in drug susceptibility was observed in drug-sensitive parasites, indicating an important role of AQP1 in the selection of artesunate resistance in *Leishmania*. AQP1 has been reported to be involved in the uptake of antimonial drugs and its downregulated expression has been found to be associated with drug resistance [80,81]. Interestingly, in artesunate resistance, higher expression of AQP1 both at the mRNA and protein levels was observed to be associated with drug resistance. Another interesting observation was the decrease in the susceptibility of drug-sensitive parasites towards artesunate at the intracellular amastigote stage, whereas no significant alteration in drug susceptibility was observed with drug-resistant parasites.

Higher expression of ABC transporters has been widely reported in drug resistance in *Leishmania* [24,81,82,83]. Upregulated expression of ABCG1 (ABCG subfamily) was observed in artesunate resistance. Further, the use of the ABC transporter verapamil resulted in a significant increase in the susceptibility of K133AS-R parasites towards artesunate both at the promastigote and amastigote stage, suggesting an important role of ABCG1 in the selection of drug resistance. However, functional characterization of ABCG1 needs to be carried out in order to establish its role in artesunate resistance. LPG5B (UDP-galactose transporter) plays diverse roles in parasite survival, like the control of parasite binding to the sand fly midgut wall, resistance to lysis by complement, protection from oxidative damage, and delayed fusion of phagolysosomes. Upregulated expression of LPG5B in K133AS-R may be helpful to parasites for their survival under drug pressure.

Comparative genome as well as transcriptome data analysis resulted in several major findings, such as upregulation of cathepsin L-like protease, amastin-like surface protein, and amino acid transporter at both the genome as well as the RNA level. Downregulated genes that were observed to be in sync with NGS data were ABCG2, Pteridine receptor, receptor-type adenylatecyclase, phosphoaceylglucosamine mutase-like protein, and certain hypothetical proteins.

Our data explicate a better insight in genomic and transcriptomics alteration that occurs during artemisinin stress under in vitro conditions and would act as a baseline for further studies involving the applicability of genomic changes encountered in the study of the clinically resistant and sensitive *L. donovani* isolated from patients of leishmaniasis. Overall, this study highlights genes and interlinked pathways contributing to artemisinin resistance using *Leishmania* as a model and highlights putative mechanisms that have applicability not only to malaria but also other diseases against which the drug is found to be effective.

## Figures and Tables

**Figure 1 genes-11-01362-f001:**
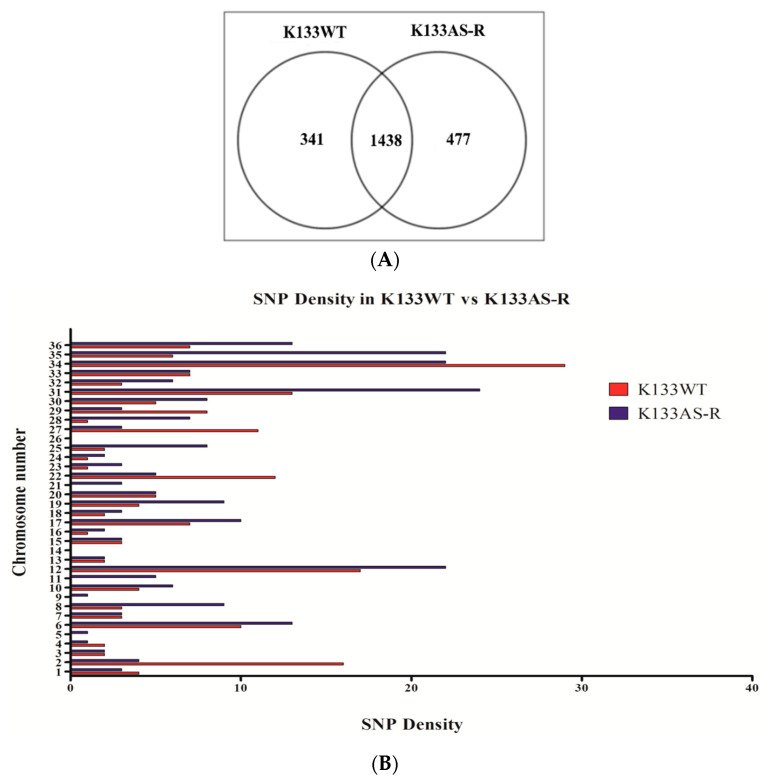
Comparative Analysis of Single Nucleotide Polymorphisms (SNPs) present in K133AS-R with K133WT. (**A**) Venn diagram showing the unique genes present in K133WT and K133AS-R. (**B**) Comparative SNP density analysis of K133WT vs. K133AS-R (**C**) Pie chart showing the percentage of different gene variants present in K133WT and K133AS-R.

**Figure 2 genes-11-01362-f002:**
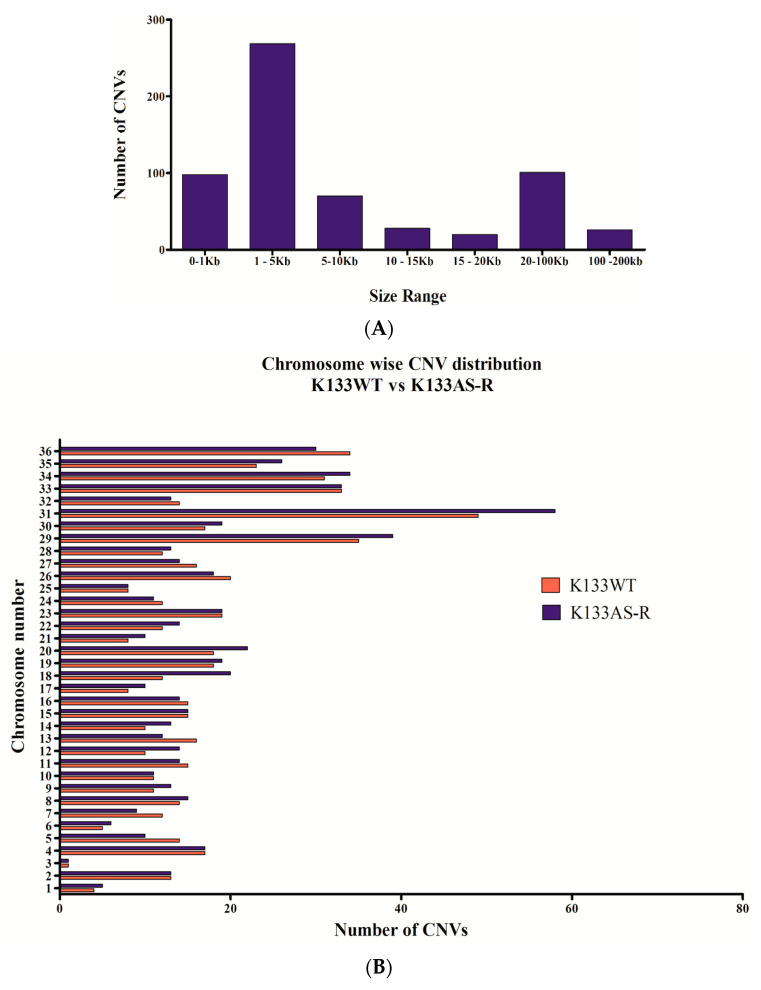
Analysis of CNV diversity in artesunate-resistant *L.donovani* (K133AS-R). (**A**) Size distribution of CNVs detected in the K133AS-R genome (**B**) Comparative CNV analysis of K133WT vs. K133AS-R.

**Figure 3 genes-11-01362-f003:**
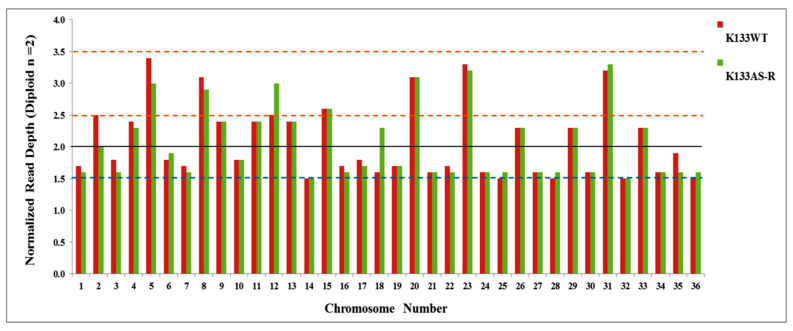
Chromosomy estimation in *L. donovani* parental line (K133WT) and artemisinin-resistant lines K133AS-R. The solid line represents median coverage and it was assigned a value of 2, considering that diploid is the principal ploidy state in *Leishmania*. The dotted line represents the calculated values for other somies (blue- monosomy; between two dotted red-trisomy).

**Figure 4 genes-11-01362-f004:**
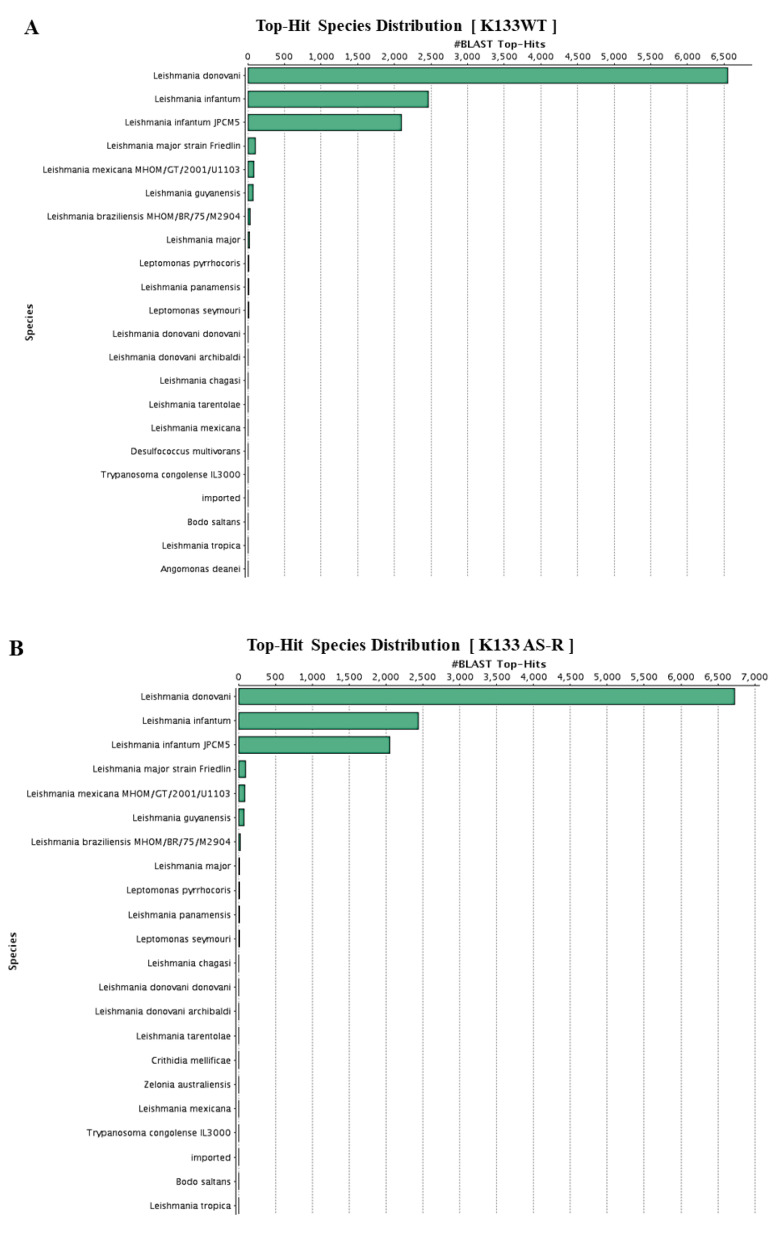
Characterization of K133WT and K133AS-R unigenes based on an NCBI non redundant (Nr) protein database search. (**A**) Species distribution of the top Blast hits for the K133WT assembled unigenes and (**B**) Species distribution of the top Blast hits for the K133AS-R assembled unigenes with a cutoff E-value of 10^−05^. Gene Ontology (GO) annotation for all the assembled unigenes in K133WT (**C**) and K133AS-R (**D**) GO-terms were assigned to functionally annotate the genes based on BLAST search results using the Blast2GO program (Biobam BioInformatics, Valencia, Spain). The results were classified based in three functional categories, Green bar represents biological function (BF); Blue: molecular function (MF); and Yellow: cellular component (CC).

**Figure 5 genes-11-01362-f005:**
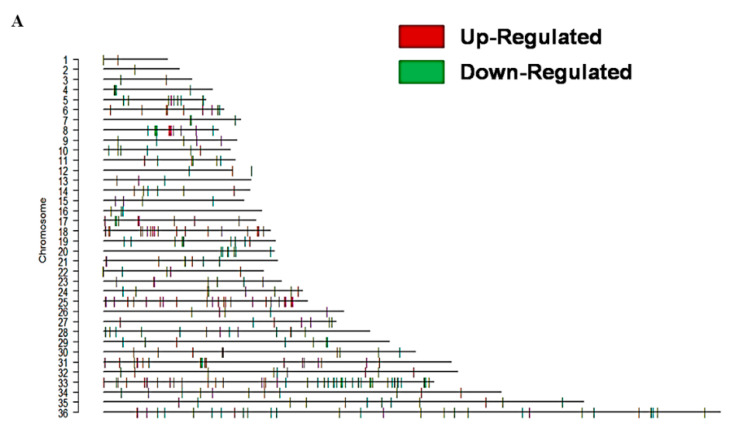
Comparative transcriptome profiling of K133WT and K133AS-R isolate. (**A**) Comparative gene expression of K133WT vs. K133AS-R parasites on the chromosome map. Chromosome map for differential gene expression was generated using Custom R program. Red lines indicate upregulated genes whereas green lines indicate downregulated genes in the K133AS-R parasite. (**B**) Percentage of differentially expressed genes in K133AS-R parasites. The percentage of modulated genes was calculated from the total 9170 genes obtained in Quality Control (QC) after filtering. Overall, 1.11% of genes were upregulated (red) whereas 1.15% of genes were downregulated (green); however, 97.74% of genes remained unaltered in K133AS-R parasites. (**C**) Categorization of genes showing differential expression in K133AS-R parasites according to GO functional categories. GO categories of differentially expressed genes in K133AS-R parasites suggested that genes belonging to various functional categories, such as metabolic processes, oxidation-reduction, cell membrane proteins, stress proteins, transporter activity, cell movement, and cell signaling, showed modulated expression. Unclassified proteins included hypothetical proteins with unknown function (that have not been characterized experimentally).

**Figure 6 genes-11-01362-f006:**
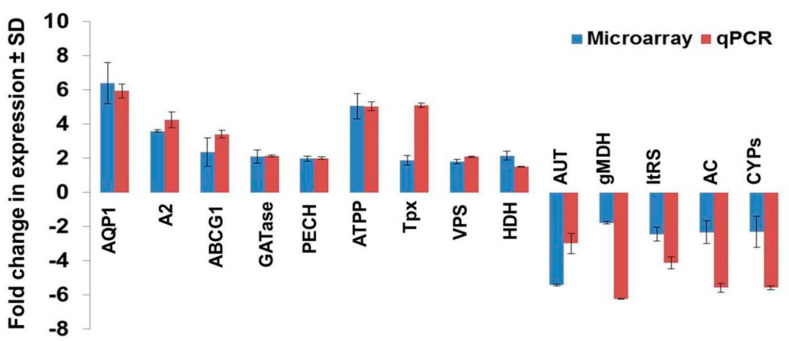
Validation of modulated expression of selected genes by qPCR. Selected 14 genes showing modulated expression in a microarray were validated for their altered expression by q-PCR in three independent RNA preparations. Fold changes in the gene expression of K133AS-R parasites with respect to K133WT parasites ± SD, obtained by q-PCR and microarray experiments, are represented here. The q-PCR data were normalized using two endogenous controls, glyceraldehyde 3-phosphate dehydrogenase (GAPDH) and cystathionine β-synthase (CBS).

**Figure 7 genes-11-01362-f007:**
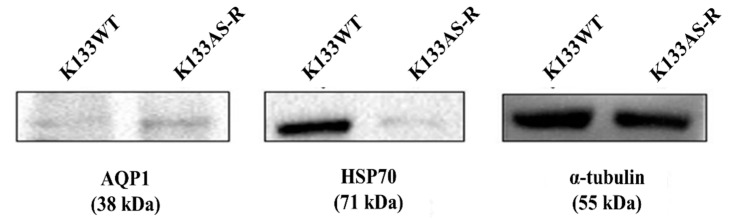
Expression analysis of AQP1 and HSP70 by Western blotting. Western blot analysis for the expression of AQP1 and HSP70 and α tubulin (endogenous control) protein was performed using 100 μgpromastigote cell lysates of K133WT and K133AS-R parasites. Proteins separated on a 12% SDS–PAGE gel, transferred to nitrocellulose membranes that were probed with anti-AQP1, anti-HSP70, or anti-α tubulin antibody followed by horseradish peroxidase (HRP)-conjugated antibody and developed using enhanced chemiluminescence (ECL).

**Figure 8 genes-11-01362-f008:**
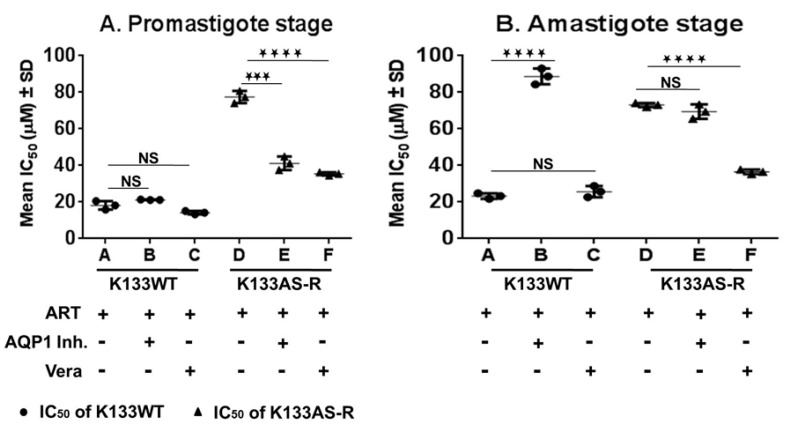
Susceptibility of K133WT/K133AS-R isolates in the presence of the AQP1 inhibitor or the modulator to ABC transporter, verapamil. In vitro susceptibility of the sensitive wild-type strain K133 WT/artemisinin-resistant strain K133AS-R isolates towards artesunate in the presence of the AQP1 inhibitor (AQP1 Inh.) and verapamil (Vera) at (**A**) the promastigote stage and (**B**) amastigote stage. IC_50_ ± SD of three independent experiments in duplicates is represented here. *** represents *p* ≤ 0.001, **** represents *p* ≤ 0.0001, NS represents not significant, Circle represents IC_50_ of K133WT, Triangle represents IC_50_ of K133AS-R.

**Figure 9 genes-11-01362-f009:**
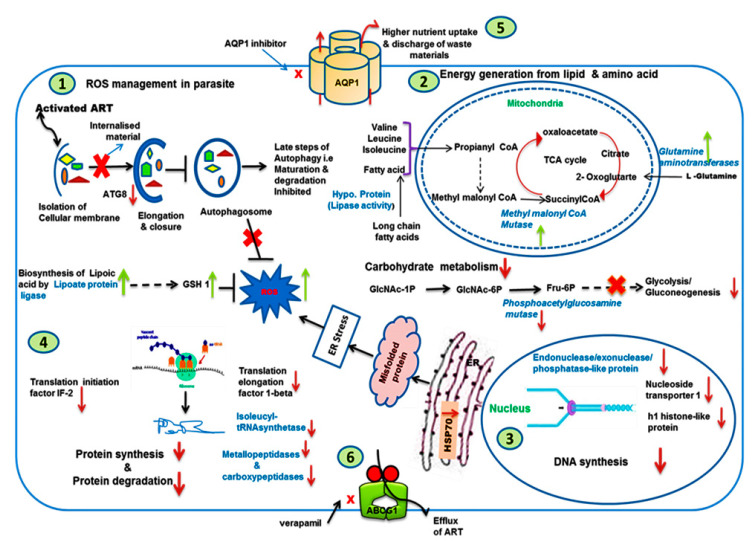
Transcriptome predicted adaptations contributing artesunate resistance in *L. donovani*: Genes altered in K133AS-R parasites are represented here. Genes marked with an up and down arrow represent, respectively, the upregulated genes and the downregulated genes in K133AS-R parasites. 1, 2, 3, 4, 5, and 6 are probable adaptations in K133AS-R parasites. (1.) Downregulation of Atg8 and HSP70 leads to increased ROS production, which was compensated by upregulation in the expression of GSH1, (2.) Upregulated expression of enzymes involved in amino acid and lipid metabolism and downregulated expression of the enzyme involved in carbohydrate metabolism, suggesting a dependency on these metabolites for energy generation, (3.) Reduced DNA synthesis that leads to the parasites in the quiescence state may be responsible for artesunate resistance in *Leishmania*, (4.) Reduced protein synthesis and reduced protein degradation, (5.) Upregulated expression of AQP1 leads to higher nutrient uptake and increased discharge of waste material and metabolic end product from the parasites and (6.) Upregulated expression of ABC transporter (ABCG1) and partial reversion or resistance in the presence of the ABC transporter modulator verapamil suggested probable involvement of the ABC transporter in the efflux of artesunate drug.

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
