# Peer review of "Genomic and Transcriptomic Analysis for Identification of Genes and Interlinked Pathways Mediating Artemisinin Resistance in Leishmania donovani"

_genes, 2020, doi:10.3390/genes11111362_

Round 1
Reviewer 1 Report
This manuscript contrasts an anteminisin-resistant strain with a sensitive one of Leishmania donovani by genomic and transcriptomic analyses. Some of the finding by these methods were further confirmed by quantitative PCR for transcripts and Western blot for proteins. In general, the manuscript is logically laid out and well written. The summary figure 10 is really helpful for readers to understand the major discovery of this manuscript. A few points need to be taken cared of to improve the manuscript.
First, SEM is used in the manuscript, for instance in figure 6, 8 and 9. SD (standard deviation) is more suitable for those cases.
Second, the section 3.8 should be combined with 3.9 and the figure 8 be presented in a complimentary figure. The data are not significant enough for its own section. These are just to show host murine macrophages responses to inhibitors, which paves way for subsequent description of parasite’s responses to these inhibitors.
Third, the text in lines 278-81 seem more logically to join the context in the lines 295-98.
Fourth, abbreviation use. You seemed to define come abbreviation in each section of the manuscript such as introduction, M&M, results and discussion, which is redundant and confusing. Once an abbreviation is first defined in the main body of a manuscript it can be used afterwards. However, this does not apply to abstract and figure legends and tables. So, some abbreviations need to be defined in figure legends.
Fifth, section 3.2. Why some words are underlined and are different is size?
Sixth, the sentences in lines 390-74 is confusing, and needs to be clearly rewritten.
Seventh, figure 9. Clearly mark which one is sensitive and which one resistant strain In the legend or figure itself.
Eighth, consider change title “Genome and transcriptome analysis for” to “Genomic and transcriptomic analyses for”
Ninth, In M&M: what was the rationale to use a chromosomal map of L. infantum rather than one of L. donovani (line 189)? What is the promastigote stage for total RNA extraction (line 206)? Primer concentration (line 211) - use molar concentration, please. This is the first time I have ever seen primer concentration in ng/ml, which is pretty much useless.
Minor points:
L36: 97 countries with 700,000 to one million
L95: from bone marrow aspirates of a VL patient
L106: add parasite stages used for gDNA extraction
The paragraph in lines 299-307. Be consistent in dismals.
Line 435: Figure 5B should be Figure 5C.
Both “anthleishmanial” and “anti-leishmanial” are used. Be consistent.
The first words of some protein names are unnecessarily capitalized such as in lines 650-51.
Line 698: “Our data serves to” to “Our data serve to”.
Author Response
Reviewer 1:
General Comment: This manuscript contrasts an artemisinin-resistant strain with a sensitive one of Leishmania donovani by genomic and transcriptomic analyses. Some of the finding by these methods were further confirmed by quantitative PCR for transcripts and Western blot for proteins. In general, the manuscript is logically laid out and well written. The summary figure 10 is really helpful for readers to understand the major discovery of this manuscript. A few points need to be taken cared of to improve the manuscript.
Response: We thank the reviewer for critical assessment of the manuscript. We have incorporated the change as suggested by the reviewers.
A list of suggestions to improve the manuscript is:
Query 1: SEM is used in the manuscript, for instance in figure 6, 8 and 9. SD (standard deviation) is more suitable for those cases.
Response 1: Figure 6 and 9 (now fig 8 as fig 8 shifted to supplementary data) have been replaced with new figures showing SD values in place of SEM.
Query 2: The section 3.8 should be combined with 3.9 and the figure 8 be presented in a complimentary figure. The data are not significant enough for its own section. These are just to show host murine macrophages responses to inhibitors, which paves way for subsequent description of parasite’s responses to these inhibitors.
Response 2: As suggested section 3.8 has been merged with section 3.9 and Figure 8 have been shifted to supplementary section and numbered as figure S2. Figure 9 is now figure 8 in main text.
Query 3: The text in lines 278-81 seem more logically to join the context in the lines 295-98
Response 3: Necessary modifications have been done as suggested.
Query 4: Abbreviation use. You seemed to define come abbreviation in each section of the manuscript such as introduction, M&M, results and discussion, which is redundant and confusing. Once an abbreviation is first defined in the main body of a manuscript it can be used afterwards. However, this does not apply to abstract and figure legends and tables. So, some abbreviations need to be defined in figure legends.
Response 4: We thank the reviewer for bringing this point to our notice. Necessary modifications have been done as suggested to maintain the homogeneity throughout the manuscript.
Query 5: Section 3.2. Why some words are underlined and are different is size?
Response 5: The words have been formatted to normal font size and underline is removed.
Query 6: The sentences in lines 390-74 is confusing, and needs to be clearly rewritten
Response 6: Lines are not clearly matching with final submitted version of manuscript. Line 370 to 374 represent GO term annotation in K133 WT vs K133 AS-R parasites.
Query 7: Figure 9. Clearly mark which one is sensitive and which one resistant strain in the legend or figure itself.
Response 7: Sensitive and resistant strains have been indicated in figure legend of fig 9 (now fig 8) for proper demarcation
Query 8: consider change title “Genome and transcriptome analysis for” to “Genomic and transcriptomic analyses for”
Response 8: Title of manuscript modified as suggested.
Query 9: In M&M: what was the rationale to use a chromosomal map of L. infantum rather than one of L. donovani (line 189)?
Response 9: Authors are thankful to the reviewer for highlighting this point. Since oligonucleotide array slide has already been printed and marketed by Agilent technologies was procured from them and used in this study. We did not customize the slide for L. donovani and purchase. Further, L. infantum is very closely related to L. donovani. So, there should not be any probable variations in the results. The similar slide was used and reported in our previous studies (Kulshrestha et al; 2014, Verma et al; 2017).
Query 10: What is the promastigote stage for total RNA extraction (line 206)?
Response 10: Early log phase promastigotes were used for total RNA extraction for both microarray and qPCR experiments. In our previous study, we reported that early log phase parasites were highly susceptible to artesunate (Verma et al; 2019). That is why parasites at this stage were used for the present study.
Query 11: Primer concentration (line 211) - use molar concentration, please. This is the first time I have ever seen primer concentration in ng/ml, which is pretty much useless.
Response 11: As suggested, primer concentration has been expressed in molar concentration and concentration of primer used was 6 pmol.
Query 12: L36: 97 countries with 700,000 to one million
Response 12: Modified as suggested
Query 13: L95: from bone marrow aspirates of a VL patient
Response 13: Modified as suggested.
Query 14: L106: add parasite stages used for gDNA extraction
Response 14: Promastigote stage of parasite (K133WT and K133AS-R) was used for genomic DNA isolation. Modification have been done at appropriate position in manuscript (Line no.)
Query 15: The paragraph in lines 299-307. Be consistent in dismals.
Response 15: As suggested necessary modifications have been done
Query 16: Line 435: Figure 5B should be Figure 5C.
Response 16: The figure is intended to be written as 5B only because this figure is representing the differentially expressed genes whereas figure 5C shows the different functional categories of genes.
Query 17: Both “anthleishmanial” and “anti-leishmanial” are used. Be consistent.
Response 17: As suggested necessary modifications have been done. The word “anti-leishmanial” has been replaced with “antileishmanial” throughout the manuscript to maintain homogeneity.
Query 18: The first words of some protein names are unnecessarily capitalized such as in lines 650-51.
Response 18: As suggested necessary changes have been done. Protein names have been rectified to normal lowercase font.
Query 19: Line 698: “Our data serves to” to “Our data serve to”.
Response 19: Sentence does not exist throughout the manuscript instead the sentence “Our data explicate” lies at Line no 680
Reviewer 2 Report
Congrats to the authors for the good work and paper,
my comments are in the attached file.

Author Response
Query 1: In several points along all the paper there are too many spaces between words.
Response 1: Necessary spacing correction have been done
Query 2: Line 42 there is a full stop before a comma
Response 2: full stop has been removed before comma
Query 3: Line 46: cut s after derivatives
Response 3: Modification done as suggested
Query 4: Line 151: a space before bracket is necessary
Response 4: Necessary incorporation done
Query 5: Line 152: a space after bracket is necessary
Response 5: Necessary incorporation done
Query 6: Line 159: at the end there are 2 full stops
Response 6: correction done as suggested
Query 7: Line 199: Is not better The data are accessible than The data is?
Response 7:Line 200: Sentence is modified now.
Query 8: Line 213: A comma needs after genes, in the end of the line
Response 8: Necessary incorporation done
Query 9: Line 296: was has to be replaced with were
Response 9: Modification done as suggested
Query 10: Line 311: ABCA7 has larger character number as well as P-type H+-ATPase, line 313-314
Response 10: Words corrected to normal font size to maintain homogeneity.
Query 11: Line 347: at the beginning, put space between do and not Same line: in ref. 38 the method is not called 2 -loop and I think this term is not correct
Response 11: Necessary spacing correction done.
We have used the similar 2-loop method as reported in previous studies [Mondelaers, et al; 2016 and Camacho, Esther, et al; 2019 (additional reference Line no……)].
Query 12: Line 363: no of genes has to be changed with n° of genes
Response 12: Modification done as suggested
Query 13: Line 389: in the reference of fig. 4 leave one space after (D)
Response 13: Space provided as per suggestion
Query 14: Line 465: in the reference of fig. 7. Put that were in place of that was
Response 14: Modification done as suggested
Query 15: In reference to Fig. 9 you need to put the significance of circles and triangles
Response 15: Significance of symbols (circles and triangles) have been incorporated in figure legend and figure 8, earlier which was figure 9. Circles represents IC50 of K133WT isolates, Triangle represents IC50 of K133AS-R isolates.
Query 16: Line 557 Is there an a to cut before ATP11?! Or is lacking a comma??
Response 16: As suggested necessary modifications have been done. a has been removed before ATP 11
Query 17: Line 558 Eliminate the term fragment?! Or put into brackets as in the line after
Response 17: Modified as suggested
Query 18: Line 572: change modulated into modulating, anyway it is better to change the term since you repeat modulated in the following line
Response 18: Modified as suggested
Query 19: Line 586 Put were in place of was
Response 19: Necessary incorporation done
Query 20: Line 641 At this point of the discussion you describe AMA1 and cite only the reference 74, which is not sufficient. I think you have to add references such as these 2:
- Meta Gene 2 (2014) 782–798 Comparative in-silico genome analysis of Leishmania (Leishmania) donovani: A step towards its species specificity of Satheesh Kumar S., Gokulasuriyan R.K., Monidipa Ghosh
- Parasitology Research (2019) 118:1609–1623 Detection and characterization of an albumin-like protein in Leishmania donovani of Bhakti Laha & Amit Kumar Verma & Bapi Biswas & Satheesh Kumar Sengodan & Akanksha Rastogi & Belinda Willard & Monidipa Ghosh
Response 20: Reference added as suggested (Ref No. 75 and 76).
Query 21: Line 681: explicate in place of explicates
Response 21: Modification done as suggested